# The Adaptive Immune System in Multiple Sclerosis: An Estrogen-Mediated Point of View

**DOI:** 10.3390/cells8101280

**Published:** 2019-10-19

**Authors:** Alessandro Maglione, Simona Rolla, Stefania Federica De Mercanti, Santina Cutrupi, Marinella Clerico

**Affiliations:** Department of Clinical and Biological Sciences, University of Turin, 10043 Orbassano, Italy; alessandro.maglione@unito.it (A.M.); simona.rolla@unito.it (S.R.); stefania.demercanti@unito.it (S.F.D.M.); santina.cutrupi@unito.it (S.C.)

**Keywords:** multiple sclerosis, adaptive immune system, epigenome, pregnancy, estrogens, estrogen receptors

## Abstract

Multiple sclerosis (MS) is a chronic central nervous system inflammatory disease that leads to demyelination and neurodegeneration. The third trimester of pregnancy, which is characterized by high levels of estrogens, has been shown to be associated with reduced relapse rates compared with the rates before pregnancy. These effects could be related to the anti-inflammatory properties of estrogens, which orchestrate the reshuffling of the immune system toward immunotolerance to allow for fetal growth. The action of these hormones is mediated by the transcriptional regulation activity of estrogen receptors (ERs). Estrogen levels and ER expression define a specific balance of immune cell types. In this review, we explore the role of estradiol (E2) and ERs in the adaptive immune system, with a focus on estrogen-mediated cellular, molecular, and epigenetic mechanisms related to immune tolerance and neuroprotection in MS. The epigenome dynamics of immune systems are described as key molecular mechanisms that act on the regulation of immune cell identity. This is a completely unexplored field, suggesting a future path for more extensive research on estrogen-induced coregulatory complexes and molecular circuitry as targets for therapeutics in MS.

## 1. Introduction

Multiple sclerosis (MS) is a chronic inflammatory demyelinating disease of the central nervous system (CNS), and is characterized by the infiltration of T lymphocytes, B lymphocytes, macrophages, and natural killer (NK) cells, as well as demyelination and axonal damage [1]. From experimental evidence based on murine models and samples from MS patients, the immunological process in the pathogenesis of MS is currently defined by the following steps. First, autoreactive T cells and B cells are activated in peripheral lymph nodes and differentiate into effector cells. Among the effector CD4+ T cells, T helper 1 (Th1) and especially Th17 cells play important roles in the pathogenesis of this disease. Patients with MS have shown increased numbers of these subpopulations in the peripheral blood [2] and the CNS, mainly in the cerebrospinal fluid and the perivascular space [3,4,5,6]. Activated T and B cells migrate through the blood–brain barrier, the disruption of which characterizes early stages of the disease, and reach the CNS, where they are further activated by local antigen-presenting cells. In the CNS, macrophages and activated CD4+ and CD8+ T cells attack myelin components and produce cytokines and chemokines that recruit other autoreactive cells from peripheral blood. They also activate B cells, which mature to antibody-producing plasma cells; induce, maintain, and reactivate CD4+ T cells; and produce proinflammatory cytokines. Overall, these processes increase inflammation and cause demyelination and axonal damage. In the advanced stages of the disease, the inflammatory response is replaced by microglial activation and chronic neurodegeneration [7].

Relapsing-remitting MS (RRMS) is the most common course of MS, and the majority of patients are initially diagnosed with RRMS. RRMS is characterized by the alternation of relapses and remissions. After RRMS, most patients transition to a secondary progressive course with the progressive accumulation of disability [8]. The onset, clinical course, and progression of MS are highly variable and likely depend on both genetic and environmental factors. However, MS does not show any clear mode of inheritance, although there is an association between first-degree relatives of patients with multiple sclerosis [9] and between twins, with the association more common in monozygotic than heterozygotic twins [10]. Among the predisposing genetic factors, human leukocyte antigens (HLA) in the class II region, especially the HLA-DRB1*1501 and DQB1*0602 alleles, have been shown to be significantly associated with MS [11], whereas the HLA class I region HLA-A*02:01 was associated with a protective effect [12]. Environmental factors, including Epstein–Barr virus (EBV) infection [13,14], smoking [15], and vitamin D deficiency [16], are known to exert epigenetic changes and have been linked to the risk of MS. More recently, evidence has suggested that other environmental risk factors for MS include intestinal microbiota [17] and oral contraceptive therapy [18]. To date, the causes that influence the development and course of MS are still not clear. 

Sex hormones could be one of the factors contributing to MS and could explain the sex inequality observed in this disease. The proposed role of sex hormones in MS is based on different clinical observations [19]. Epidemiological studies have shown differences in the prevalence and progression of the disease between men and women. The relapsing forms of MS are more frequent in young women [20]; the disease in men usually develops at an older age, with a more severe and progressive course, possibly in relation to an incipient decline in androgen secretion [21]. Moreover, the relapse rate decreases during late pregnancy as hormonal secretions increase [22]. This was first shown in 1998 by the Pregnancy in Multiple Sclerosis (PRIMS) study, which prospectively assessed 254 women with MS during pregnancy and reported a 70% reduction in the annualized relapse rate in the third trimester compared with the rate in the year before pregnancy [23,24]. A meta-analysis [25] that included 1221 pregnancies in women with MS showed a significant decrease in the relapse rate during pregnancy. Moreover, these results were supported by a larger multicenter retrospective study [26]. Furthermore, in MS patients, estradiol regulates immune responses by regulating the expression and release of inflammatory and anti-inflammatory cytokines, leading to a regulatory immune response [27]. These data suggest a potential role of estrogens in MS, although few clinical trials have been completed so far.

In the following sections, we discuss the current knowledge of the relationship between estrogens, the immune system, and MS from a cellular, molecular, and epigenetic point of view. 

## 2. Estrogens

Estrogens are sex steroid hormones that are present in both men and women, but they circulate at significantly higher levels in women during reproductive age. Endogenous estrogens include estrone (E1), 17β-estradiol (E2), and estriol (E3). E2 is the predominant form of estrogen in premenopausal women, while E3 is mainly produced during pregnancy, together with E2. The level of circulating estrogens varies during all stages of a woman’s life, starting from childhood until menopause (Figure 1). In each of these stages, hormonal changes have fixed influences on the female body. Estrogens primarily promote the development of female secondary sexual characteristics and regulate the menstrual cycle. In addition to sexual development, estradiol influences the functionality of various organs and tissues, including the skin, muscles, adipose tissue, the brain, the cardiovascular system, and bones, and it actively protects against osteoporosis and various cardiovascular diseases [19]. Even the immune system is affected by the levels of circulating estrogens, especially during pregnancy, when it reacts adaptively to establish fetal tolerance [28]. In physiological pregnancy, maternal regulatory T (Treg) cells expand in both the periphery [29,30] and the placenta, in which they suppress the aggressive allogeneic response directed against the fetus. The lack of Treg cells leads to failure of gestation due to the immunological rejection of the fetus [31,32]. Treg cells also have the ability to suppress autoimmune responses. Indeed, the protective effects of estrogens in MS are believed to partly result from a combination of estrogen-mediated anti-inflammatory cytokine production and Treg cell expansion [29,32,33]. During menopause, the risk of autoimmune diseases increases in women. Older women show a stronger pro-inflammatory response compared to males due to the senescence of the immune system that leads to a sex-specific low-grade chronic inflammation [34]. The decline of estrogens’ concentration during menopause correlates with a reduced number of B and T cells and an increased secretion of pro-inflammatory cytokines [35]. 

### Estrogen Receptors 

Estrogens act directly, indirectly, or both, and their mode of action depends on the involvement of their receptors, called estrogen receptors (ERs) [37]. ERs are nuclear steroid receptors that are able to dimerize upon activation and translocate to the nucleus, where they regulate gene expression. Activated ERs can bind directly to specific DNA sequences called estrogen response elements (EREs) and act as transcription factors (TFs) by regulating a broad range of estrogen-responsive genes. Alternatively, ERs can indirectly bind DNA through protein–protein interactions with other transcription factors [38,39]. In the absence of estrogens, ERs have been shown to bind extensively to the genome of breast cancer cells and regulate the expression of hundreds of genes with developmental functions [40].

ERs exist in two main forms, ER alpha (ERα) and ER beta (ERβ), which are encoded by the human genes Estrogen Receptor 1 (*ESR1*)and Estrogen Receptor 2 (*ESR2*). ERα and ERβ share high homology, particularly in the DNA binding domain [41]. The general structure of ERα consists of an N-terminal activation function-1 domain (AF-1), which is followed by a DNA binding domain (DBD), a dimerization domain, and the ligand binding (LB)/AF-2 domain. The AF domains are responsible for the recruitment of coregulators; cofactor recruitment by AF-1 is ligand-independent, whereas cofactor recruitment by AF-2 is ligand-dependent. [42]. Three main different isoforms of ERα, derived from alternative splicing events, have been described: the full-length 66 kDa ERα (ERα66), the AF-1 domain-truncated 46 kDa variant of ERα (ERα46), and a 36 kDa ERα variant (ERα36) that lacks both AF-1 and AF-2 domains [43,44,45] (Figure 2). Similarly, ERβ is transcribed from at least two additional upstream promoters and undergoes alternative splicing, leading to at least five protein isoforms (ERβ1–5) [41]. 

The ovary, uterus, and breasts express ERs in abundance and, therefore, represent the main target tissues of estrogens. However, estrogens affect many other tissues, including the immune system, in which ER signaling contributes to the regulation of the immune response.

Along with gene expression, a fundamental aspect of ER function in the cell is the recruitment of coregulating proteins that are necessary for mediating the transcriptional activity of ERs. The resulting complexes contribute to epigenetic modifications and chromatin remodeling that transform the response to hormones or pharmacological ligands involved in regulatory activity [46]. Epigenetic modifications are hereditary modifications that do not alter the DNA sequence but regulate gene expression. At the DNA level, the most frequent epigenetic modification is the methylation of cytosine in CpG islands. Usually, hypomethylated CpG islands are associated with active genes, while CpG hypermethylation tends to silence gene expression. At the chromatin level, on the other hand, histone acetylation and methylation model chromatin and form active regulatory regions as enhancers and promoters or repressed heterochromatic regions (e.g., histone H3 lysine 27 acetylation in active regulatory regions increases the accessibility of chromatin to TFs). DNA methylation and demethylation often contribute to the inheritable organization of chromatin, while histone modifications are able to confer cellular identity but remain sufficiently malleable to regulate response to stimuli. Once activated, ERs recruit chromatin remodeling complexes in a timed and sequential manner. These mechanisms have been described in detail in the model MCF-7 breast cancer cell line [46,47]. The described ERα-associated transcriptional coactivator complexes include histone arginine methyltransferases (e.g. p160, CARM1), histone acetyltransferases (e.g. CBP/p300, TIP60, GCN5), RNA-processing factors (e.g. SRA), and polymerase II mediator complexes (e.g. TRAP/DRIP/ARC). Conversely, corepressors include chromatin remodeling complexes (e.g. SWI, NURD) and basal corepressors with histone deacetylase activity (e.g. NCoR, SMRT). The Next-generation sequencing (NGS) technologies have broadened the understanding of these processes by showing estrogen binding to ERs in distal regulatory regions to modulate the expression of several hundreds of target genes [48,49]. In recent years, the recruitment of coregulators has been shown to lead to the remodeling of chromatin’s three-dimensional (3D) organization [50]. This 3D rearrangement results in the formation of functional chromatin loops between ERα binding sites at the enhancers and promoters of target genes that are activated [51,52,53]. The formation of loops mediated by ERα is also involved in the mechanisms of gene repression. Estrogen-mediated DNA looping represses diverse chromosomal regions through DNA methylation and repressive chromatin modifications that inhibit gene expression [54]. Furthermore, ERα activity is influenced by the tissue-specific presence of coactivators and transcriptional corepressors and their differential interaction with ERα in the presence of estrogens or anti-estrogens [55,56].

## 3. Estrogen Effects on the Immune System: Focus on MS

Increasing evidence is continuing to highlight the action of estrogens on the immune system. These aspects have been described in both physiological (e.g., pregnancy) and pathological conditions of the immune system (e.g., autoimmunity and the tumor microenvironment [57,58]). 

Changes in circulating estrogen levels can affect progenitor and mature cells of both the innate and adaptive immune systems. ERα is present in most cells from the early stages of hematopoietic development to lymphocyte development in the thymus [57,59,60]. In the early stages, E2 enhances the expansion of hematopoietic pluripotent stem cells (hPSCs) [60], the differentiation of monocytes to macrophages [61], thymus trophism, and the maturation of double positive cells (CD4+ CD8+) [62,63] through ERα-dependent pathways. 

The first illustration of estrogenic effects on the immune system emerged from the analysis of ER expression in peripheral blood mononuclear cells (PBMCs). ER expression has been explored by using different techniques: quantitative TaqMan RT-PCR analyses, flow cytometry, and Western blotting have indicated that ERs are differentially expressed in PBMC subsets [64,65]. 

Gene expression analysis by quantitative PCR (qPCR) has shown that ERα and ERβ are endogenously expressed in Th lymphocytes [64], and their expression levels in B lymphocytes seem to be higher than those expressed in CD4+ T cells and CD8+ T cells; this is especially the case for ERβ. Comparisons between CD4+ T cells and CD8+ T cells suggest that CD4+ T lymphocytes express higher levels of ERα. The immunostaining approach has been used to confirm and better characterize the expression of a specific receptor in the same cell type. This approach has shown that CD4+ and CD8+ T lymphocytes, B lymphocytes, and NK cells contain intracellular ERα and ERβ, and data suggest that ERβ is expressed at a lower level with respect to ERα [65]. Interestingly, the short isoform ERα46 is the most represented isoform in T cells compared with ERα66 [65]. The ERα46 protein is also predominantly expressed by human macrophages in addition to the full-length ERα66 [61] (Figure 2). ERα46 is formed by skipping exon 1, which encodes the AF-1 domain that is responsible for ligand-independent transactivation. ERα46 and ERα66 share a ligand-binding site and a DNA binding site, but they differ in the AF-1 domain. As a result of this difference, the mechanisms of coregulator recruitment differ between cells with high levels of the short isoform and target tissues in which the long isoform predominates and is constitutionally expressed at very high levels. Specific tissue mechanisms depend on expression, the heterodimerization of receptor isoforms, competition for DNA binding sites, or a combination of these processes [43]. Moreover, ERα66 and ERα46 have similar estrogen binding affinity, but they bind differentially to some estrogen receptor agonists and antagonists. In particular, a classical estrogen receptor antagonist, ICI 182,780 (Fulvestrant), was found to have a higher affinity for ERα66 than ERα46 [66]. In the age of NGS, gene expression databases are a popularly used tool to explore cell-type-specific gene expression levels. Figure 3 shows the gene expression level of ER receptors in immune system cells. Interestingly, the expression of ERα and ERβ is greater in all B and T lymphocyte subtypes and NK cells in the non-activated state compared with in vitro activated lymphocytes and circulating monocytes.

### 3.1. Innate Immune Cells 

The role of ERs in the regulation of innate immune system cells has been described in recent reviews [59,68,69,70], which have suggested estrogens’ potential contribution to sex differences in the innate immune response by affecting both progenitor and mature cells.

The role of ERs in the regulation of the development and functions of innate immune cells has been discussed in detail in other reviews and is beyond the purpose of our manuscript [20,57,58]. However, we report the main findings (summarized in Table 1), especially those linked with MS. Estrogens affect the innate immune system by regulating the number of cells and their specific biological functions: in neutrophils, they regulate chemotaxis, infiltration, and the induction of cytokine-induced neutrophil chemoattractants (e.g., CINC-1, CINC-2β, CINC-3) and cytokines (e.g., TNF-α, IL-6, IL-1β); in macrophages, they regulate chemotaxis, phagocytic activity, and the production of cytokines (e.g., IL-6, TNF-α); in NK cells, they decrease cytotoxicity; in dendritic cells (DCs), they promote differentiation and regulate chemokine (e.g., IL-8 and chemokine C-C motif ligand 2 (CCL2)) and cytokine (e.g., IL-6, IL-10) expression [20,57]. 

In the context of MS, ERα activation delays the onset of experimental autoimmune encephalomyelitis (EAE), while ERβ activation sustains later neuroprotection. Indeed, both ERα and ERβ signaling reduce demyelination, axonal loss, and neuronal pathology in EAE, but only ERβ activation induces the recovery of motor performance [71]. The anti-inflammatory action of ERα is connected to the modulation of microglia, which survey the CNS for infections and have functions that are similar to macrophages in the periphery [72]. ERα regulates the inflammatory pathway in microglia, likely by reducing the time of nuclear factor kappa-light-chain-enhancer of activated B cells (NF-κB) transcriptional activity and thus regulating inflammatory signaling [73,74]. The later neuroprotection mediated by ERβ activation is connected to the observed effects on macrophages in the CNS. ERβ activation induces CD11c+ DCs and macrophages to express less inducible NOS (iNOS) and T-box transcription factor TBX21 (T-bet) and more IL-10, and these effects favor immunotolerance in EAE mice. Furthermore, ERβ activation induces the maturation of oligodendrocytes and enhances remyelination [75]. The innate and adaptive immune systems are closely connected, and it has become evident that estrogens can regulate the interactions among immune cell types. Indeed, ERs sustain neuroprotection in EAE by regulating the interactions between innate immune cells and both T [76] and B cells [77]. 

### 3.2. T Cells 

Estrogens can act on the adaptive immune system by modulating the production of cytokines and interleukins and influencing the differentiation of lymphocytes and the inflammatory environment (summarized in Table 1). 

E2 modulates cytokine secretion by CD4+ T cells from healthy subjects and self-reactive CD4+ T cell clones isolated from MS patients. Low concentrations of E2 (i.e., levels during the pre-ovulatory phase of the menstrual cycle) induce IFN-γ production in T cells in mice [78,79], humans [80], and MS Th clones [81]. IFN-γ is the principal cytokine secreted by activated T cells as well as other cell types, such as NK, B, and antigen-presenting cells (APCs), in order to promote cell-mediated immunity. IFN-γ stimulation by estrogens is mediated by ERα regulation of the IFN-γ gene [78], the Th1-specific transcription factor T-bet [82], or both. On the other hand, high doses of E2 (i.e., levels during pregnancy) in these immune cells induce the expression of the transforming growth factor beta (TGF-β) and anti-inflammatory interleukin 10 (IL-10) [81,83]. Although E2 is able to stimulate both IFN-γ and IL-10 at the same time, the results of these two events do not seem to conflict. An increase in the concentration of estradiol favors immunotolerance by significantly decreasing the IFN-γ/IL-10 ratio [84]. Moreover, in human CD4+ T cells, the production and secretion of TNF-α were seen to increase at low E2 concentrations and be inhibited at high E2 concentrations [81].

Estrogens have a less marked effect on IL-4 production in CD4+ T cells [81,83,85]. IL-4 antagonizes the effects of IFN-γ and thus inhibits T cell-mediated immunity. During the menstrual cycle, a positive correlation exists between estrogen levels and IL-4 [86]. The hormone progesterone induces IL-4 production in Th cells [87] but does not affect IL-12, IFN-γ, IL-10, and TNF-α [84]. During pregnancy, the modulation of IL-4 is attributed to progesterone, and the immune-tolerance environment can be realized and maintained by the combined action of progesterone and estrogen, which affect the synthesis of various anti-inflammatory cytokines [88]. The effects of estrogen on cytokine regulation in adaptive immune cells are summarized in Figure 4.

Estrogens at pregnancy levels enhance the expression of the transcription factor forkhead box P3 (FOXP3), which is specific for Treg, in mice [89]. We recently demonstrated that FOXP3 expression is promoted in human PBMCs upon stimulation with pregnancy levels of estradiol from Th17 cells undergoing polarization in vitro [90]. Moreover, estradiol potentiates the suppressive function of Treg cells by promoting their proliferation [91]. Estrogens also regulate immune checkpoints. Immune checkpoints involve proteins that modulate the signaling pathways responsible for immunological tolerance. Programmed cell death protein 1 (PD-1) and cytotoxic T lymphocyte-associated protein 4 (CTLA-4) are immune checkpoint proteins, and their expression is regulated by ERα-mediated signaling [92,93]. The anti-inflammatory effect of estrogens also involves Th17 cells. Th17 cells, which are characterized by the production of the proinflammatory cytokine IL-17, have been associated with the pathogenesis and outcome of several autoimmune diseases, including MS [2,94]. Moreover, estrogen deficiency in postmenopausal women is associated with increased IL-17A levels [95]. 

The importance of estrogens in the modulation of the adaptive immune system during MS is supported by data from the EAE murine model of MS. In mice with EAE, pregnancy limits cell infiltration and reduces CNS demyelination. Induced immunization during pregnancy leads to a reduction in the incidence of EAE and a decrease in clinical severity, while immunization during the postpartum period increases the severity of the disease [96]. In addition, the effects of pregnancy are evident even when the pregnancy occurs after the onset of EAE [97]. The protective effect is mediated by a reduction in TNF-α- [98] and IL-17-secreting cells and an increase in IL-10-secreting cells. E2 promotes immune tolerance by enhancing the Treg cell compartment and FOXP3 expression [89]. E2 treatment in mice strengthens the expression of PD-1 in Treg cells in a dose-dependent manner and correlates with the efficiency of EAE protection. E2 at pregnancy levels, but not at lower concentrations, increases the frequency of Treg cells and drastically reduces the production of IL-17 in the peripheral blood of immunized EAE mice. Treatment with E2 does not protect against EAE in mice with PD-1 deficiency [99]. Moreover, *Esr1* -/- immunized mice are not protected against EAE in the presence of E2. The splenocytes of *Esr1* -/- mice produce more TNF-α, IFN-γ, and IL-6, even in the presence of E2. In contrast, in wild-type (WT) mice and *Esr2* -/- mice, E2 treatment produces clinical signs of EAE suppression and eliminates inflammatory lesions in the CNS [100]. These results show that the reduction in EAE severity involves the genomic action of E2 via ERα [71] and that the anti-inflammatory effect is mediated by ERα but not ERβ [71,100]. Moreover, experiments using ERα-deficient mice have demonstrated that T lymphocytes (but not macrophages or dendritic cells) require ERα for the E2-mediated inhibition of Th1/Th17 cell differentiation and protection from EAE [101]. The results of these studies emphasize the role of Th17 and Treg cells in ERα-mediated E2 modulation in EAE. 

### 3.3. B Cells 

Estrogens also have profound effects on B cell maturation [102], differentiation, activity [103,104], and survival [105]. Estrogen has been shown to increase the numbers of plasma cells and autoantibody-producing cells [103]. Estrogens promote IL-10 secretion in regulatory B cells (Breg), a specific subset of B cells that can negatively regulate T cell immune responses, thereby controlling the follicular T cell response in germinal centers [106]. Together with Treg cells, the frequency of Breg cells increases during pregnancy [107].

B cells contribute to the pathogenesis of MS by producing anti-myelin antibodies, acting as antigen-presenting cells, and producing cytokines [108,109]. Interestingly, recent evidence has demonstrated that B cells are required for E2-mediated protection against EAE. The effects of E2 on Breg cells are mediated through ERα and the PD-1 pathway. Treatment with E2 upregulates PD-L1 in B cells and increases the percentage of Breg cells that produce IL-10. These results suggest that the anti-inflammatory effects of estrogens are also mediated by Breg cells, which suppress neuroinflammation during EAE and reduce the number of proinflammatory cells that infiltrate the CNS [110,111,112].

## 4. Estrogens Modulate the T Helper Epigenome in MS

The specific genomic regulatory landscape of cells controls gene expression and defines cell identity. The phenotypes of Th cells are determined by their cytokine secretion, gene expression, and surface molecules, which guide their action in the adaptive immune system. Th cells can react to changes in environmental stimuli by repolarizing to different cell subtypes in a phenomenon defined as plasticity [128]. Epigenetic reprogramming is a series of events that underlie plasticity, and this process determines the difference between a pro-inflammatory and an anti-inflammatory environment [129]. In this context, chromatin functions as a device that controls the immune response. As previously discussed, methylation of DNA contributes more to the stable organization of chromatin, while histone modifications can regulate transitory responses to stimuli. Histone modifications are able to maintain a stable cellular state while remaining sufficiently malleable to allow for plasticity in Th cells. In fact, the histone modifications that determine the accessibility of chromatin to TFs can change in response to different situations and stimuli [130]. One of the pioneering studies on this subject described changes in histone modification at the promoter of lineage-determining TFs in T cells as a molecular mechanism that occurs during cell plasticity [131]. Considerable data depict a more complex molecular mechanism in which distal genomic regulatory regions, such as enhancers, become active after the binding of TF complexes [50]. Epigenome dynamics in T cells have been described and discussed, starting from their development in the thymus to their peripheral plasticity [132].

The balance between Th17 and Treg is widely considered to reflect inflammation in MS and is strongly connected to disease outcomes [133]. Th17 and Treg have a high degree of plasticity, which allows for their functional adaptation to the phases of the immune response. However, Th17-Treg plasticity could also be a critical factor in MS [134]. The integration of genome-wide data on the regulation of the epigenome and transcriptome by TFs has helped to unravel the intricate gene regulatory circuits underlying these processes in Treg [90,135] and Th17 cells [90,136,137]. Some epigenetic regulation mechanisms and targets have been associated with EAE and the Th17–Treg axis. In encephalitogenic T cells of EAE mice, signaling through CD44 causes increased methylation of *Ifng/Il17a* and demethylation of *Il4/Foxp3* [138]. Since CD44 expression is chronically elevated in MS demyelinating lesions, this mechanism has been proposed to sustain inflammation at the sites of CNS lesions [138]. Conversely, the CD27 and CD70 costimulatory pathway results in the epigenetic silencing of the IL17a gene, thus inhibiting Th17 differentiation [139].

In particular, FOXP3, given its role as a key transcription factor in Treg cells, has long been studied in the context of epigenetic regulation and autoimmunity. The demethylation of the conserved non-coding sequence (CNS0) in the *FOXP3* locus helps to stabilize the identity of Treg cells [140]. In addition to CNS0, at least two other known CNSs are responsible for *FOXP3* regulation (i.e., CNS1 and CNS2) [141]. Recent studies on CNS1—a *FOXP3* intronic enhancer that is essential for the development of peripheral Treg cells—have reported that the adaptation of the immune system during pregnancy enabled maternal–fetal tolerance [140]. Moreover, the deletion of CNS2—a *FOXP3* enhancer—led to reduced stability and the loss of *FOXP3* expression in proliferating Treg cells [140,142,143]. However, FOXP3 alone does not control all aspects of Treg biology and is not the initiating factor in Treg development. DNA demethylation of Treg signature genes is required for the stable maintenance of the Treg phenotype and function [144,145]. The establishment of the Treg-specific epigenome starts before FOXP3 expression. Indeed, FOXP3 exploits a pre-existing enhancer landscape and a TF network of Treg cells [146,147,148]. Ten-eleven translocation (TET) proteins regulate DNA methylation and gene expression by converting 5-methylcytosine (5mC) to 5-hydroxymethylcytosine (5hmC). Treg cells in mice with specific Tet2/Tet3 deficiency begin to express IL-17. This phenotypic shift occurs not only at the level of known CNSs but also in new regions identified as *FOXP3*’s upstream enhancer, which could contribute to stable FOXP3 expression [149]. DNA methyltransferase 3A (DNMT3A), responsible for “de novo” methylation, prevents methylation of the *FOXP3* locus [150], thus supporting Treg cell identity at sites of inflammation by keeping CNS2 in a demethylated state and allowing for the maintenance of its suppressive function. Interestingly, the epigenetic reprogramming of peripheral Treg cells is possible to achieve in vitro through the demethylation of the RAR-related orphan receptor C (*RORC*) locus and the development of Th17-like cells [151].

The role of estrogens and ER in the complexity of epigenetic regulation mechanisms in T cells has been poorly studied, but some evidence has emerged from recent studies. As previously described, estrogens promote the activation of ERα and its transcriptional activity through interactions with ERE. ERα binding at the *RORC* and *FOXP3* regulatory regions has been recently demonstrated. In both in vitro experiments and pregnant MS patients, E2 at pregnancy levels inhibited Th17 polarization, thereby reducing *RORC* expression and enhancing *FOXP3* transcription as a result of ERα binding to their promoters and enhancers (Figure 5) [90]. The molecular mechanisms of this process remain elusive. However, the suppressive action of ERα in Th17 cells could be mediated by the recruitment of the repressor of estrogen receptor activity (REA). The ERα/REA complex recruits histone deacetylases to the *RORC* promoter to suppress its expression [152]. In the orchestration of chromatin architecture, ERα may mediate epigenetic modifications at chromatin hubs in CD4+ T cells to influence their differentiation and plasticity (Figure 5). In this respect, ERα may act as a cooperative TF in T cell epigenome dynamics. Understanding the steps that lead to this mechanism may open doors to new therapeutic approaches that exploit this property of T cells.

## 5. Estrogens as a Potential MS Therapy

To mimic the protective effects of estrogens observed during pregnancy, E3 was administered in 10 female MS patients for 6 months. Then, the treatment was discontinued for the next 6 months, followed by 4 months of retreatment. The results showed that the number and volume of MRI lesions decreased in all patients; this clinical observation was correlated with reduced IFN-γ levels [153]. MRI-enhancing MS lesions increased 3 months after treatment was stopped. In parallel, in vitro analysis showed reduced production of TNF-α and the upregulation of anti-inflammatory IL-5 in CD4+ T lymphocytes and IL-10 in macrophages [153]. Estriol was well tolerated, no serious side effects were observed; there were neither significant alterations in any laboratory measures including sexual hormone levels [153]. These promising preliminary results led to a larger phase 2 trial [154] that enrolled 164 female MS patients, who were treated with glatiramer acetate (GA), an immunomodulating drug used as a first line therapy for MS, and E3 as an add-on therapy compared with GA alone. The results showed a reduced annualized relapse rate in the group treated with E3 (0.25 relapses per year group versus 0.37 relapses per year in the control group). Moreover, serum E3 concentrations were inversely correlated with the number of relapses and the number of active lesions on brain MRI [154]. No differences were observed in the number of cerebral lesions (enhancing or T2 lesions) [154]. Safety analysis showed that the serious adverse events proportion did not differ between the treatment and the placebo group. Post-hoc MRI investigations using volumetry study, showed at 12 months less cortical grey matter atrophy in the estriol group than in the control group [154]. 

In 2009, a double-blind placebo-controlled phase 3 study enrolled 300 pregnant MS women [155]. The main objective of the study was to prevent postpartum MS relapses by treatment with nomegestrol acetate (oral administration) 1 day after delivery and an E2 patch 2 weeks later (i.e., 15 days after delivery). The results did not show any beneficial effect on relapse rates or on the MRI [155]. A more recent phase 2 clinical trial was conducted in female MS patients who received high-dose ethinylestradiol and desogestrel in addition to IFN-β, an immunosuppressant drug widely used as first-line treatment in relapsing forms of MS. The results showed a significant decrease in new gadolinium-enhancing lesions compared with patients who received only IFN-β over a 96-week period [156]. In this study no serious adverse events were detected [156].

As occurs in the postpartum period, the estrogen drop during menopause could favor MS relapses [157]. A retrospective questionnaire-based study on menopausal and premenopausal women with MS [158] showed that 82% of menopausal women reported that the severity of their symptoms worsened during the premenstrual period. Among postmenopausal women, 54% reported worsening symptoms after menopause, and 75% of postmenopausal women who tried hormone replacement therapy reported disease improvement [158].

## 6. Conclusions

Estrogens regulate immune cell responses and exert anti-inflammatory and neuroprotective effects in MS. Because the differences in the immune cell pattern are maximized during pregnancy, the pregnant condition represents a model for investigating the immunological changes that determine protection from disease activity.

In this review, we describe the phenotypical changes in adaptive immune cells induced by estrogens, spanning from ER expression in different immune cell types, through estrogen-induced cytokine modulation, to estrogen-induced epigenetic changes in T cells. A complex picture can be depicted in which ERs may interact with genomic regulatory regions to recruit coregulators and chromatin remodelers. The NGS approach could pave the way to a deeper knowledge of the gene expression and regulation involved in these cell dynamics, as shown in recent studies that have explored MS brain lesions at single-cell resolution [159,160,161].

Although challenging, identifying the molecular mechanisms that underlie pregnancy-induced protection from relapses in MS could lead to potential therapeutic targets to apply as alternative or complementary treatments to those already used. Estrogen, as a therapeutic tool, needs further investigation in addition to the ongoing phase 3 study. These studies are essential for the development of new therapeutic opportunities to prevent postpartum relapses or for the treatment of women with persistently high disease activity who are generally advised to delay pregnancy [162]. Furthermore, the use of a molecular target could optimize the effectiveness of estrogen to avoid its side effects.

## Figures and Tables

**Figure 1 cells-08-01280-f001:**
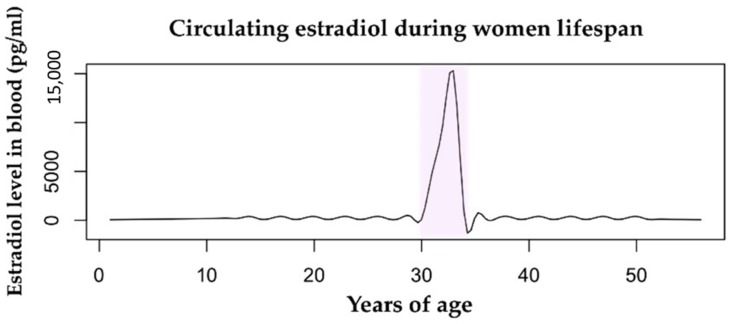
Estradiol levels in the bloodstream vary throughout a woman’s lifespan. The mean value during childhood is 200 pg/mL. During fertility age, the menstrual cycle range is 100–400 pg/mL. The pregnancy condition (highlighted in pink) is characterized by a huge increase in levels of circulating estradiol from the first trimester to delivery, with a range of 2000–15,000 pg/mL. During menopause, the level of estrogens drops drastically to <100 pg/mL. Data retrieved from Watson et al., 2010 [36].

**Figure 2 cells-08-01280-f002:**
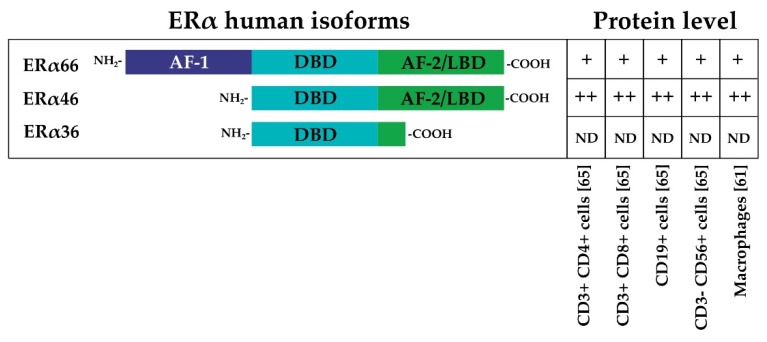
The three main different isoforms of ERα are presented: the full-length 66 kDa ERα (ERα66), the AF-1 domain-truncated 46 kDa variant of ERα (ERα46), and a 36 kDa ERα variant (ERα36) that lacks both AF-1 and AF-2 domains [43,44,45]. The relative protein levels in different immune system cells are indicated by ++, +, or ND (not detected) [61,65].

**Figure 3 cells-08-01280-f003:**
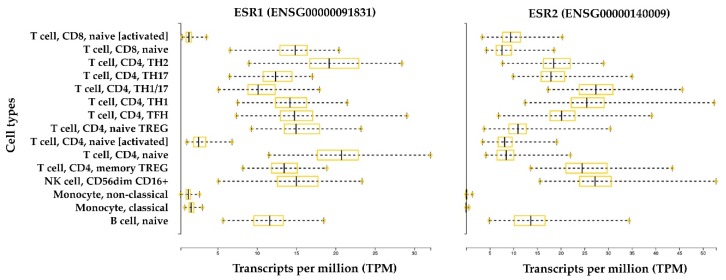
ERα and ERβ expression in the immune system. The bar plots represent gene expression data of the human genes *ESR1* and *ESR2*, which encode for ERα and ERβ, respectively. Data were retrieved from the Database of Immune Cell expression, expression quantitative trait loci (eQTL), and epigenomics (DICE) [67]. RNA-Seq data are normalized between samples and expressed in transcripts per million (TPM). Data were generated from 13 immune cell types from 91 healthy subjects. The cell types include: three innate immune cell types (CD14high CD16− classical monocytes, CD14− CD16+ non-classical monocytes, and CD56dim CD16+ natural killer (NK) cells); four adaptive immune cell types that have not encountered their cognate antigen in the periphery (naive B cells, naive CD4+ T cells, naive CD8+ T cells, and naive Treg cells); six differentiated T cell subsets (Th1, Th1/17, Th17, Th2, follicular helper T cells (TFH), and memory Treg cells); and two ex vivo activated cell types (naive CD4+ and CD8+ T cells).

**Figure 4 cells-08-01280-f004:**
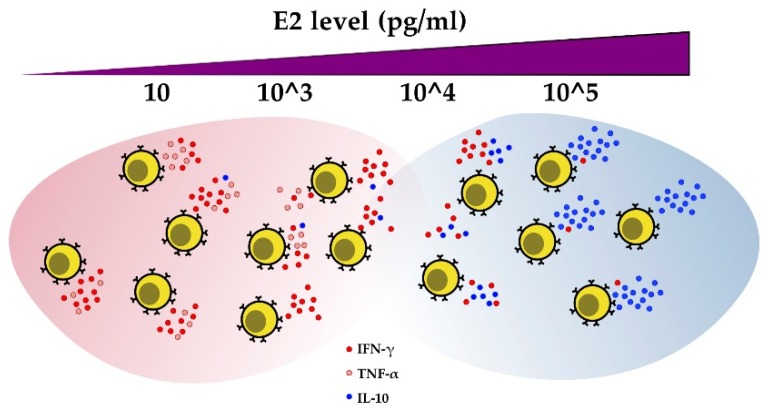
E2 regulates cytokine production in CD4+ T cells. As estrogen levels increase, IFN-γ and TNF-α production decreases, while IL-10 secretion increases.

**Figure 5 cells-08-01280-f005:**
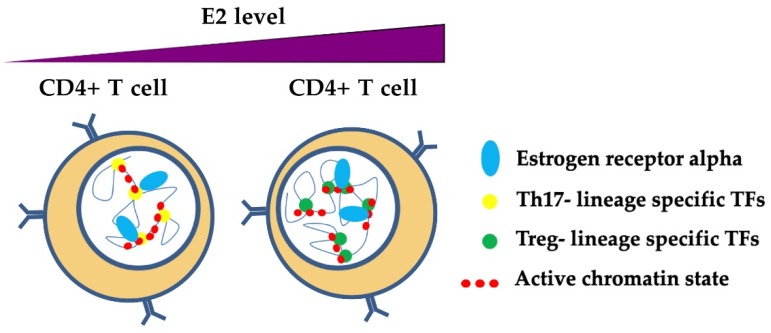
A model of ERα-dependent modulation of nuclear organization of chromatin in T helper cells. Estrogens participate in the mechanisms of transcriptional regulation through the binding of ERα at regulatory regions, thereby influencing the phenotype of T helper cells. Estrogens at normal levels promote the binding and activation of Th17 lineage-specific TFs (e.g., *RORC*), whereas estrogens at pregnancy levels bind preferentially to Treg lineage-specific TFs, thus inhibiting RORC and promoting *FOXP3* transcriptional activation [90]. ERα may participate with TFs that are specific for Th17 and Treg lineages in chromatin remodeling in these cells, although the mechanisms are still unclear.

**Table 1 cells-08-01280-t001:** Effect of estrogens on different immune cell types.

Cell Type	Effect in Immune System	References	Effect in EAE/MS	References
Neutrophils	↓ TNF-α, IL-6, IL-1β	[113]		
	↓ Chemotaxis (iNOS, CINC-1, CINC-2β, CINC-3)	[114]		
Macrophages	↓ TNF-α, IL-6, IL-1β	[115]		
	↓ iNOS and T-bet	[75]	↓ iNOS and T-bet	[75]
	↑ IL-10	[75]	↑ IL-10	[75]
Dendritic Cells	↑ Differentiation (IL-8, and CCL2)	[116,117]	↑ Differentiation (IL-8, and CCL2)	[116]
	↓ iNOS, T-bet, TNF-α, IFN-γ, IL-12, PD-L1, PD-L2	[75,116,117,118]	↓ iNOS, T-bet, TNF-α, IFN-γ, IL-12, PD-L1, PD-L2	[75,116,118]
	↑ IL-10	[75,117]	↑ IL-10	[75]
	↑ IL-6, IFN-γ	[119]		
Microglia	↑ M2 polarization	[77,120]	↑ M2 polarization	[77]
	↓ Activation	[76]	↓ Activation	[76]
	↓ NF-kB, IL-1β	[73]		
	↑ IL-10	[73]		
NK	↓ Cytotoxic activity	[121]		
	↑ Activation of CD3+CD56+CD8+ cells	[122]	↑ Activation of CD3+CD56+CD8+ cells	[122]
T cells	↑ Treg/Th2	[120,123]	↑ Treg/Th2	[123,124]
	↓ Th17/Th1	[123,125]	↓ Th17/Th1, T cell infiltration in CNS	[123,125]
	↓ TFH cell response	[126]	↓ TFH cell response	[126]
	↓ T CD8+ cells	[120]		
	↑↓ IFN-γ	[78,79,80,81,82,83]	↓ IFN-γ	[79,80,81,83,127]
	↑↓ TNF-α	[81,98,123]	↓ TNF-α	[81,98,123]
	↑ IL-10, IL-4, TGF-β	[83,84]	↑ IL-10, IL-4, TGF-β	[83]
	↓ IL-17 and IL-23	[123]	↓ IL-17 and IL-23	[123,127]
	↓ NF-kB, iNOS	[76]	↓ NF-kB, iNOS	[76]
	↑ PD-1, CTLA4, FOXP3, GATA3	[83,90,92,93]	↑ PD-1, CTLA4, FOXP3, GATA3	[83,90,92,93]
	↓ RORC, T-bet	[90,123]	↓ RORC, T-bet	[90,123]
B cells	↑ IL-10	[77]	↑ IL-10	[77]
	↑ PD-L1	[110,111,125]	↑ PD-L1	[110,111,125]

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
