# Peer review of "The Adaptive Immune System in Multiple Sclerosis: An Estrogen-Mediated Point of View"

_cells, 2019, doi:10.3390/cells8101280_

Round 1

Reviewer 1 Report

The article by Maglione et al, is not well written and is very hard to understand the presented information.  It requires considerable understanding of multiple sclerosis before reading the article.  Consider introducing interferon and glatiramer acetatate as well as CD4+ and CD8+ cells before mentioning them. Please consider english assistance with this article.  The described work deserves to be published but it is very difficult to understand as it is currently written.

Reviewer 2 Report

I red the review of Maglione and colleagues on the role of estrogens in MS. While I think this is an interesting and clinically relevant topic, I think that this manuscript needs major revisions before I could recommend publication. I have three main issues with the manuscript

1.) First and foremost, it seems very unstructured – you jump between different topics back and forth (just to mention one example, in section 3 and 4 you jump between different immune cell types and their relation to ER or E). This made it very difficult for me to perceive your messages. One possible (likely not the only) approach would be: start the review with briefly introducing MS as presumable auto-immune disorder and the potential role of hormones in etiopathogenesis. Then introduce E (and potentially also its receptors?). Then elaborate on the relation between MS and E/ER, try to keep a structure here: E in innate immunity, E in adaptive immunity, effect on which immune cells. Independent how you do it, pls improve the structure of your manuscript.

2.) Along point 1: a figure/schematic (e.g. with the different immune cells and the effect of E on them) and/or a table (eg. listing the immune cells and the effect of E on them) summarizing the findings would help the reader to get the underlying mechanisms. Such a schematic would also increase citation numbers.

3.) This manuscripts needs in-depth proof-reading for English grammar and spelling. Just to name a few issues:

First paragraph: women elderly – should be elder women

Figure legend 1: During menopause the level of estrogens drops down drastically – should be level of estrogen drop. And “pregnancy condition” seems an odd term to me.

“Increasing evidences highlight that the action…” – there is no plural for evidence.

“Changes in circulating estrogens level…” – circulating estrogen levels.

“since increasing estradiol concentration IFN-γ/IL-10 ratio decreases significantly” – I do not understand this sentence.

Further specific comments:

The whole review should use the word sex rather than gender, because it clearly relates to the biological differences between men and women and not between social/cultural differences.

“Even the immune system is influenced by levels of circulating estrogens and reacts adapting to what is necessary to face: i.e. during pregnancy maternal immune system adapt to establish the fetal tolerance” – pls add reference.

“Elder women show more pro-inflammatory response as compared to males. Male body undergoes less significant changes also, but these occur mainly at the level of estrogen-regulated pathways.” – pls add reference.

HLADRB1*1501 is not the only susceptibility haplotype in MS – there is also HLA-A*02.

“numbers of brain lesions measured by Magnetic Resonance Imaging (MRI) and Gadolinium contrast media” – what type of brain lesions? T1 black holes? T2 hyperintensities? I would rather state gadolinium enhancing lesions than just Gd contrast media.

“The observed annualized relapse rate was lower in the estriol group compared to the placebo group with no observations of adverse events in estrogen-related carcinogenic action” – add numbers in brackets.

“number of cerebral lesions” – pls specify

You argue that the phase 3 estriol trial is important to introduce new therapeutics for MS in pregnancy – but phase 1 – 3 are in non-pregnant women apparently; it also seems inappropriate to treat pregnant women with additional estriol.

“Moreover, TNF-α is the principle cytokine that mediates acute inflammation” – this is just one part of the truth; effect of TNF highly depends on its receptor (TNFR1 or R2) and can also be anti-inflammatory (see negative effect of anti-TNF-antibody in MS).

“maternal regulatory T (Treg) cells expand both in the periphery [80, 81] and locally” – what do you mean by locally?

Pls introduce EAE abbreviation.

The last paragraph from the conclusion seems very odd and I do not really get the message it tries to convey.

You briefly mention DMD in pregnancy and rebound after giving birth. What is the evidence for E – does it play a role? Is there anything known? Pls elaborate.

Reviewer 3 Report

Maglione et al. review the literature on estrogens axis on adaptive immunity during multiple sclerosis (MS). The topic is of high interest to the filed of autoimmunity as pregnancies have been shown to improve and sometimes reverse autoimmune diseases. How exactly estrogens interact with the immune axis, remain poorly understood.
The review is well constructed and clearly written; it goes over the critical research finding in the field.
It starts with a general overview, and then goes into results of clinical trials and presents a multitude of possible molecular mechanisms.
Minor points to improve
This review mostly concentrated on the relatively old papers with only a couple of reports younger than 2014. I would encourage authors to incorporate in this review most recent finding.
Also, it would be helpful if authors would state in the introduction how their review is different from many others.
Interestingly most of the research revolves around E2, while a few clinical trials used E3. Is there any particular reason for these discrepancies?
Finally, the theme of estrogenic side effect deserves its own section in the review.

Author Response

POINT BY POINT REPLAY TO REVIEWER 3

This review mostly concentrated on the relatively old papers with only a couple of reports younger than 2014. I would encourage authors to incorporate in this review most recent finding.

We implemented references with other recent papers as suggested.

Also, it would be helpful if authors would state in the introduction how their review is different from many others.

The main differences form other reviews consist in the discussion of the current knowledge about the relationship between estrogens, immune system and MS from a cellular, molecular and epigenetic point of view. This has been reported in several ways in the abstract, in the introduction, and also in the conclusion.

Interestingly most of the research revolves around E2, while a few clinical trials used E3. Is there any particular reason for these discrepancies?

Different experiments have been performed in the murine model that indifferently use both E2 and E3 (Lim et al. Neurology 1999, 12;52:1230), even if the model currently recognized is that of Polanczyk et al., (100), mainly for two reason: i) E2 has a higher binding affinity to ERs compared to E3; ii) E3 was less active than E2 subcutaneously in mice. In patients, clinical trials are currently ongoing both E2 (155, 156) and E3 (153, 154) and we reported them in the text.

Finally, the theme of estrogenic side effect deserves its own section in the review.

Ccertainly the theme of estrogenic side effects is important from the clinical point of view. In this regard, we added in the text the result from the POPART’MUS study and the study of Pozzilli et al., where no serious adverse events were reported.

Round 2

Reviewer 1 Report

The presented information in the revised article by Marglione et al, is easier to understand.  The presented information is worth publishing.  However, I do have a few minor comments to make about the article.

Minor comments

References needed after the following sentences: 

Line 202:" This approach has shown that CD4+ and CD8+ T 201 lymphocytes, B lymphocytes, and NK cells contain intracellular ERα and ERβ, and data suggest that 202 ERβ is expressed at a lower level with respect to ERα." Line 372: "In encephalitogenic T cells of EAE mice, signaling through CD44 causes increased methylation 374 of IFN-γ/IL-17A and demethylation of IL-4/FOXP3. "

Author Response

References needed after the following sentences: 

Line 202:" This approach has shown that CD4+ and CD8+ T 201 lymphocytes, B lymphocytes, and NK cells contain intracellular ERα and ERβ, and data suggest that 202 ERβ is expressed at a lower level with respect to ERα." Line 372: "In encephalitogenic T cells of EAE mice, signaling through CD44 causes increased methylation 374 of IFN-γ/IL-17A and demethylation of IL-4/FOXP3. " 

We added references 65 and 138 as requested.

Reviewer 2 Report

The review improved its structure and english language and is thus better readable.

Author Response

We thank the reviewer for appreciating our work